# Associations Between Daily-Use Products and Urinary Biomarkers of Endocrine-Disrupting Chemicals in Adults of Reproductive Age

**DOI:** 10.3390/ijerph22010099

**Published:** 2025-01-13

**Authors:** Jayne Marie Foley, Carol F. Kwiatkowski, Johanna R. Rochester, Iva Neveux, Shaun Dabe, Michael Kupec Lathrop, Eric J. Daza, Joseph J. Grzymski, Ben K. Greenfield, Jenna Hua

**Affiliations:** 1Million Marker Wellness, Inc., Berkeley, CA 94704, USA; jayne@millionmarker.com (J.M.F.); carol@millionmarker.com (C.F.K.); jo@millionmarker.com (J.R.R.); mike@millionmarker.com (M.K.L.); ericjdaza@gmail.com (E.J.D.); 2Healthy Nevada Project, Renown Health, Reno, NV 89557, USA; ineveux@med.unr.edu (I.N.); dabe316@gmail.com (S.D.); jgrzymski@med.unr.edu (J.J.G.); 3Department of Internal Medicine, University of Nevada, Reno, NV 89557, USA; 4Public Health Program, Muskie School of Public Service, University of Southern Maine, Portland, ME 04104, USA; benjamin.greenfield@maine.edu

**Keywords:** endocrine-disrupting chemicals, personal care products, bisphenols, phthalates, parabens

## Abstract

Background: Daily-use products, including personal care products, household products, and dietary supplements, often contain ingredients that raise concerns regarding harmful chemical exposure. Endocrine-disrupting chemicals (EDCs) found in daily-use products are associated with numerous adverse health effects. Methods: This pilot study explores the relationship between concentrations of EDCs in urine samples and products used 24 h prior to sample collection, and ingredients of concern in those products, in 140 adults of reproductive age in Northern Nevada. Results: Having higher numbers of products and ingredients of concern, especially in the personal care category, was associated with higher levels of mono-(-ethyl-5-carboxypentyl) phthalate (MECPP). Similarly, taking more supplements was associated with higher levels of methylparaben (MePB). In contrast, using household products with more ingredients of concern was associated with lower levels of monobutyl phthalate (MBP). Generally, women used more products, were exposed to more ingredients of concern and had higher urinary metabolites than men. Participants who rated themselves as being in poor/fair health were exposed to more personal care and supplement ingredients of concern than those in better health. Interestingly, those in excellent health also took supplements with more ingredients of concern. Conclusions: Greater product use and more ingredients of concern are associated with urinary metabolites of known EDCs and self-ratings of poor health. Women and people who take supplements are at greater risk, and even people who consider themselves to be healthy can be highly exposed. More education among the general public is needed to make people aware of the presence of these chemicals in their everyday products so they can make efforts to avoid them.

## 1. Introduction

Many widely used synthetic chemicals are endocrine-disrupting chemicals (EDCs), which can interfere with the endocrine system’s balance and function [1]. EDCs are found in numerous products that people are exposed to daily, such as shampoos and conditioners, lotions, sunscreens, clothing, plastics, pesticides, cookware, and food [2,3]. Over the last several decades, endocrine-disrupting chemical exposure has grown substantially [4]. Furthermore, many EDCs are present in products due to contamination during production and packaging [5,6,7]. While some EDCs have been banned, regulation is difficult because chemicals are diverse. There is no standard for reporting all known chemicals and their risks; health effects may show up months or years later, and regulatory testing is expensive and time-consuming [8,9]. In recent years, there have been numerous studies on the effects of various endocrine disruptors on human health endpoints, including reproductive issues in both men and women [10,11], diabetes [12], neurological disorders [13], and certain types of cancer [14,15]. Based on this research, many health and scientific organizations, including the American Medical Association (AMA), the World Health Organization (WHO), the Endocrine Society, and the American College of Obstetricians and Gynecologists (ACOG) have recommended counseling patients to reduce EDC exposure [16,17,18,19,20]. For example, the College of Obstetricians and Gynecologists (ACOG) recommends that doctors “screen and counsel patients during the pre-pregnancy and prenatal periods about opportunities to reduce toxic environmental health exposures”, particularly EDCs [18]. Despite these recommendations, EDC-specific environmental health literacy in healthcare providers is poor, and thus patient EDC counseling is not the norm, leaving individuals responsible for their own education [16,21].

The average U.S. adult uses 12 personal care products a day with an average of 112 unique chemical ingredients [22]. Previous studies have shown associations between personal care product use and urinary EDC biomarkers. A study of over 200 pregnant women indicated that those who avoided certain ingredients in products like “fragrance” had lower phthalate concentrations, particularly monoethyl phthalate (MEP) [23]. MEP was also found in higher concentrations in a study with a group of men who used cologne or aftershave within 48 h of collection of their urine sample for EDC testing [24]. Other research has identified a similar link between children using personal care products and elevated concentrations of phthalates [25]. A variety of interventions have shown that the use of personal care products is directly associated with urinary biomarker concentrations of EDCs. Both research interventions that instruct individuals to stop using products containing these chemicals and those that instruct individuals to begin using them (for a short time period) all found that the use of these products increased the concentration of EDCs in urine [26].

An estimated 70% of all chronic illnesses are related to an individual’s exposome (environment) and the remaining 30% are related to an individual’s genetics (genome) [27]. This suggests humans may have the ability to significantly alter their health based on their environmental exposures. While regulation at the state and federal levels and health provider support is essential for the protection of human and environmental health, the fact that so many chemicals are still in use puts the onus of reducing these chemicals on the consumer. Reporting health testing results (“report-back”) as an intervention has shown promise in increasing environmental health literacy and healthy behavior changes [28,29,30,31,32,33]. However, there are few options for individuals to understand their common exposure risks (e.g., EDCs in products, etc.) and proactively track and reduce them. Direct-to-consumer health testing (e.g., 23andMe, California, United States) as well as digital health tools and applications (e.g., Fitbit) with feedback services have become increasingly popular [34,35,36]. These technologies have been shown to increase general health literacy [37,38,39,40], but challenges exist in effective comprehension of report-back and science/clinical translation [29,41,42]. Additionally, while patients can order genetic tests to learn about their (unchangeable) health predispositions, there have previously been very limited tools/services available for the understanding of EDC exposures in order to make actionable changes to prevent disease and optimize health.

In a previous paper, we found that reporting back EDC metabolite levels to individuals, with actionable recommendations for EDC reduction, led to increased environmental health literacy and reduced EDC exposure in a population of US men and women [43]. We also found that women reported increased readiness to reduce exposures (while men reported the opposite). These results may indicate that men required more support or more tailored recommendations. Indeed, participants reported increased knowledge after the study, but also more concern about product choices and expense, indicating the need for further education and individualized support [43]. Thus, there is a need for the analysis of individuals’ product use with the aim of providing tailored recommendations to reduce exposure.

In another relevant study, researchers swapped out personal care products in a population of adolescent girls, and found a decrease in phthalate, paraben, and phenol chemical concentration in urine samples over a three-day period, showing that improving personal products can reduce EDCs [44]. However, it is unclear how the analysis of individuals’ real-world personal care/household product use relates to EDC exposure, health, and other important demographics. Thus, in this study, we compared individual exposure data (i.e., 24 h exposure journals detailing product use) and ingredients of concern within those products, as well as sociodemographic and health parameters, with urinary EDC metabolite levels, in order to better understand the potential impacts on human health, and to further the development of tailored, actionable feedback to reduce EDCs.

## 2. Materials and Methods

All study methodologies received approval from the General Institutional Review Board (IRB) at the University of Nevada, Reno, which is responsible for overseeing social, behavioral, and educational research, as well as biomedical studies and clinical trials (IRB00000215; “[1786153-1] Renown Institute for Health Innovation-Million Marker Detect and Detox Pilot”, approved on 19 July 2021). Participants did not receive compensation for their involvement.

### 2.1. Population

We recruited participants from the Healthy Nevada Project (HNP), a population health and genetics research study [45]. The recruitment process, consent procedures, and data collection for participants in the HNP are detailed in Grzymski et al. [46]. For the current study, from August 2021 to July 2022, a total of 526 recruitment emails were sent to HNP participants who agreed to be contacted about future research studies and met the age criteria for this study. Those who responded were required to meet the following eligibility criteria: be between the ages of 18 and 40, own a smartphone, speak English, not be pregnant, and have no known diagnoses of cancer, metabolic disorders, or kidney disease. During the recruitment phase for the HNP, data on participants’ age, sex, race/ethnicity, education level, income, height, weight, and self-reported health status were gathered through a questionnaire.

### 2.2. Biomarker Measurements

Participants had to complete a preliminary survey at the start of the study. Then, they received a Detect & Detox Kit from Million Marker (MM), based in Berkeley, CA, USA [47]. The kit includes a urine sample cup made of polypropylene, a return label, and packaging, as well as instructions for sample collection. Additionally, participants could access detailed instructional videos on the MM website to help minimize the risk of contamination during sample collection. Urine samples were to be collected during the first morning void. If participants completed the exposure journal, they were instructed to complete the journal referring to the 24 h prior to sample collection. Participants then sent their samples back to the third-party laboratory using 2-day shipping with FedEx Priority Overnight to ensure prompt delivery. Upon arrival at the lab, samples were immediately logged, aliquoted, and stored in a −80 °C freezer. The following metabolites were analyzed in the urine samples: bisphenol A (BPA), bisphenol S (BPS), bisphenol F (BPF), monobutyl phthalate (MBP), monoethyl phthalate (MEP), mono(EthylHexyl) phthalate (MEHP), mono-(2-ethyl-5-hydroxyhexyl) phthalate (MEHHP), mono-(2-ethyl-5-carboxypentyl) phthalate (MECPP), methylparaben (MePB), ethylparaben (EPB), propylparaben (PPB), butylparaben (BUP), and oxybenzone (OBZ). These metabolites are commonly found in the urine of more than 95% of the US population [48,49,50,51,52]. The urine samples were analyzed using liquid chromatography/tandem mass spectrometry (LC/MS/MS) after preparation. In brief, 100 μL of urine was combined with 100 μL of water, isotopically labeled standards were added to both the samples and blank water, and cocktail standards were spiked at 5 μL each. The samples and standards were incubated with 25 μL of β-Glucuronidase buffer for 2 h at 37 °C, followed by the addition of 275 μL of water to the vials. Solvent blanks were prepared simultaneously. Samples were injected in duplicate, with blanks inserted after each duplicate. The analysis was conducted using an Agilent Prochell 120-EC18 column (Agilent Technologies, Santa Clara, CA, USA) [43].

Data from individuals with metabolite concentration values less than the limit of detection (LOD) were replaced by the LOD divided by 2, as suggested for highly skewed data [53]. After applying this transformation to the relevant metabolite values, the metabolite concentration values for all metabolites were adjusted for specific gravity (SG) to obtain a urinary dilution-corrected concentration for each participant using the following formula: adjusted biomarker concentration = observed biomarker concentration × ((study sample SG Median − 1)/(individual SG − 1)). Of the 208 participants with EDC biomarker data, 17 did not receive specific gravity measurements due to an error at the lab and, thus, were removed from further biomarker analyses. Due to the extreme positive skew of the biomarker data (regardless of transformation), concentration values were divided into quartiles and treated as ordinal data for analysis. Specifically, the ordinal ranks of the values were assessed, rather than the values themselves. For several biomarkers (MEHP, MEHHP, BPA, BPF, BPS, EPB, BUP, and OBZ), fewer than 25% of participants had concentrations above the LOD. These biomarkers were removed from further analyses.

### 2.3. Exposure Data

Exposure data were gathered from participants who maintained a 24 h exposure journal to identify potential sources of exposure, including personal care products, food and beverages, household items, supplements, and lifestyle activities from the day prior to urine sample collection. Data related to food and beverage exposures were excluded from this analysis due to inconsistencies in reporting. Participants submitted their exposure journal reports via a tool available on a mobile application platform. The questions encompassed both multiple-choice and open-ended formats. Participation in journal reporting was optional, and only those who provided this information were included in the analyses for this study.

After the products reported by participants were collected through the app, the ingredients listed on the product labels were added to a database of products and ingredients. “Ingredients of concern” were identified by searching banned and restricted lists and authoritative databases with chemical hazard data such as the US Environmental Protection Agency CompTox Database [54], the California Safe Cosmetics Program Product Database [55], and the US National Library of Medicine PubChem Database [56]. Each of these databases provides hazard ratings based on dozens of other sources of information on hazardous chemicals.

### 2.4. Data Analysis

A priori hypothesis was that greater product usage and more ingredients of concern would be associated with higher levels of urinary EDC metabolites, and that those variables would differ by certain demographics. All other reported analyses were exploratory, i.e., their results should be used to generate hypotheses for testing in a subsequent study with new data. For these exploratory analyses, our priority was to discover “true positive” results, rather than protect against “false positives” as is more appropriate for confirmatory analyses. Hence, we did not adjust our statistical findings for multiple testing (i.e., multiplicity). For all analyses, the data were reviewed to ensure assumptions were not violated [57].

When the outcomes were not sufficiently normally distributed for tests to be valid, nonparametric tests were conducted to compare ordered ranks across categories. In some instances, variables with response categories with too few participants were combined to avoid violating the assumptions of the statistical tests. Zero values in product variables were set as missing values because it was not possible to determine a zero from a lack of product entries in the exposure journal.

The specific statistical tests used are footnoted under each table and described in the associated text. A *p*-value was considered statistically significant if it was <0.05, although given the exploratory nature of this pilot study, results with *p*-values < 0.1 are also described. Throughout the paper, we made efforts to use language that accurately reports “statistical significance” as indicative of the quality of statistical evidence of study findings. For example, each finding with a statistically significant *p*-value is described as being statistically discernible, supported, or reliable to avoid the misconstrual of statistical significance as indicating clinically or practically meaningful data [58,59,60,61,62,63]. Data were analyzed using the IBM SPSS Statistical Software (Version 29).

## 3. Results

Of the 191 participants who had metabolite data, 140 had product usage data from the exposure journal and were analyzed for this study.

### 3.1. Demographics

Out of the 140 participants, 82% were female. The average age of participants was 31 years. The majority of participants identified as White (81%), the remaining being Hispanic, Asian, Native American, Pacific Islander, or Black. Out of the 140 participants, 107 had additional demographics regarding height and weight, education, and health status; 106 reported their income. The average BMI of these participants was 26. Almost all of these participants had some level of higher education (96%) and about half had an annual income between USD 50,000 to USD 100,000. When asked about their current health status, the majority of respondents said they were in very good (50%), or good (31%) health. The demographics are shown in Table 1.

### 3.2. Products and Ingredients of Concern

Table 2 shows that participants had used on average 5.5 products (median) within the previous 24 h, with a maximum of 38. Most of these were personal care products, with far fewer in the household products and supplement categories. It is likely that participants underreported product usage because it is challenging to identify all the products one uses and reporting them was time-consuming. The average number of ingredients of concern was 19.5 (median), with a maximum of 233. These also mostly fell under the category of personal care products.

Figure 1 and Figure 2 below represent waterfall plots of the reported products and ingredients of concern of each individual participant, in order from left to right according to the total numbers of products and ingredients of concern. Spikes in the graph are noteworthy because they indicate a participant with a high proportion of a subcategory of products or ingredients (personal care, household, or supplements) relative to the total number of products they reported.

For the a priori analysis of demographics, the numbers of products and ingredients of concern in all categories were grouped such that values less than their respective medians were categorized as “Low”, and those greater than or equal to their respective medians were categorized as “High”. We likewise dichotomized the numbers of personal care products and ingredients of concern relative to their medians.

Table 3 shows the overall demographic distribution for the total number of products and the total number of ingredients of concern. The relationship between the BMI and the number of products reported was statistically discernible (U = 1045.5, *p* = 0.018). Individuals with a slightly lower BMI reported a higher number of products used. There was also statistical support for the association between sex and both the number of products (χ^2^ = 3.94, *p* = 0.047) and the number of ingredients of concern (χ^2^ = 3.94, *p* = 0.047). Specifically, women on average reported more products and had more ingredients of concern than did men.

Exploratory demographic analyses were also conducted for the subcategories of product and ingredient data. Similar to the overall variables, the average BMI was lower among those who reported more personal care products (U = 799, *p* = 0.048). Compared to men, women reported more household products (U = 255.5, *p* = 0.017) and supplements (U = 406.5, *p* = 0.027) on average, and perhaps also personal care products (U = 770, *p* = 0.061). Women also reported more personal care ingredients of concern (U = 760, *p* = 0.051) on average, although this is likely because they reported more personal care products.

Relationships with moderate statistical evidence (i.e., that were above the *p* < 0.05 statistical significance level but below 0.10) in these exploratory analyses included the following. On average, those with lower income (less than USD 50,000) used fewer household products than participants in higher income categories (H = 5.061, *p* = 0.08). With regard to health status, those who rated themselves as being in poor or fair health generally used personal care products with more ingredients of concern than those who rated themselves as being in good, very good, or excellent health (H = 7.47, *p* = 0.058). Interestingly, those who rated themselves as being in poor/fair health or in excellent health generally used supplements with more ingredients of concern than those in very good health (H = 6.44, *p* = 0.092).

### 3.3. Urinary Metabolites

Table 4 shows the five metabolites, with at least 25% of all participants having concentrations above the LOD. The metabolite levels were not normally distributed, with many participants below the LOD and several participants with extremely high values for one or more metabolites. Thus, we categorized the metabolites into quartiles, which groups this variable into four ordinal parts based on the 25, 50, and 75 percentile cut-offs in the data [64]. The means of each quartile are shown in Table 3.

Exploratory demographic comparisons were run for each metabolite, and results with *p*-values of 0.1 or lower are described here. Women had higher levels of MEP (U = 1059, *p* = 0.033) and MePB (U = 1105, *p* = 0.062) levels than men. For education, participants with some higher education or less had lower levels of MECPP than those with an associate or master’s degree or higher (H = 7.013, *p* = 0.071). People who rated themselves as being in poor health and excellent health had higher MePB levels (H = 17.309, *p* < 0.001) than those in good and very good health, and similarly, people in poor health had higher PPB levels (H = 6.745, *p* = 0.08) than those in good health. For MECPP (H = 11.481, *p* = 0.009) and MBP (H = 7.625, *p* = 0.054) levels, the better the health status the higher the metabolite level.

A priori analysis of the relationships between dichotomized (low/high) product usage or ingredients of concern and the five metabolites revealed the following results. As shown in Figure 3, participants with higher levels of MECPP reported using more total products (*p* = 0.013 in Table 5). Participants with higher levels of MECPP may also have used more total ingredients of concern (*p* = 0.083 in Table 5). Those with higher levels of MeBP reported using more supplement products (*p* = 0.03 in Table 5). Interestingly, the direction of the effect between the number of ingredients of concern for household products and MBP levels was the opposite of the other variables (although this was not significant at *p* < 0.05). The direction of the relationships can be seen in Figure 3, although the numbers of personal care products and ingredients of concern are not shown for MECPP because they are subcategories that follow the same pattern as the total products and ingredients of concern.

## 4. Discussion

This pilot study revealed that people who used more products, specifically more personal care products, in the 24 h prior to their urine tests had higher levels of the phthalate MECPP in their urine. This was also true for people who were exposed to more personal care ingredients of concern. MECPP is a metabolite of di-(2-ethylhexyl) phthalate (DEHP) which is a plasticizer found in polyvinyl chloride and building materials, as well as plastic packaging for food and products [6,65,66]. It is likely that the correlation of higher DEHP with increased product/ingredient count is due to DEHP contamination. This contamination can be introduced at many points in the manufacturing process, including during the production, shipping, and storage of ingredients, formulation and mixing (e.g., tubing, machinery), and leaching from final packaging. Consumers may seek out products labeled “phthalate-free” but these products often unintentionally contain phthalates due to contamination. Furthermore, many small-batch boutique products, advertised as “clean”, may have even more contamination, due to less control over the sources and storage of ingredients [5,67].

Surprisingly, participants who had higher levels of ingredients of concern from supplements had lower MBP levels in their urine. MBP is a metabolite of dibutyl phthalate (DBP). Low molecular weight phthalates such as DBP are used in personal care products as odor diluents/solvents and are not required to be disclosed on product labels. They are often included in the many ingredients that make up “fragrance” or “parfum” [68]. These low-molecular-weight phthalates are also found in the coatings of pharmaceuticals and nutritional supplements, including prenatal vitamins [3].

Participants who used more supplement products had higher levels of methylparaben (MePB). MePB is a preservative, and has long been used in foods, pharmaceuticals, and cosmetics to extend shelf life [69]. Interestingly, higher levels of MePB and using more supplements were also associated with reporting being in better health. Prior research confirms that people who take supplements are more likely to report themselves as healthy [70,71]. However, they are likely unaware that these supplements may be harming their health.

Regarding overall product use and ingredients of concern, participants reported using eight products on average, with a maximum of 38 products. After documenting the ingredients in each of these products and assessing the hazards associated with each ingredient, we determined that on average, participants were exposed to 39 ingredients of concern, with a maximum of 233. While this number may seem high, consider that a single body cream that contains 36 ingredients has 25 ingredients of concern, including several EDCs such as triethanolamine and butylated hydroxytoluene. One of the ingredients was simply listed as “fragrance” which often contains numerous ingredients of concern, including EDCs such as phthalates. A popular laundry detergent contains 65 ingredients with 52 ingredients of concern, with some of the 13 remaining ingredients having no data with which to judge their hazardous properties. This detergent also contains 39 ingredients subsumed under the term “fragrance”, which are not named on the label but are available through SmartLabel [72].

On average, participants used more personal care products than household products or supplements, and personal care products contained by far the most ingredients of concern. Women used more products and were exposed to more ingredients of concern than men, which was consistent across all three subcategories of products, and also for personal care ingredients of concern. In addition, women in our study had higher levels of MEP and MePB than men [43]. These findings support prior research [73] and the conclusion that women are at higher risk for health effects associated with exposure to EDCs.

Regarding self-reported health status, people who rated themselves as being in poor or fair health were exposed to more ingredients of concern from personal care products and supplements and had higher levels of MePB and PPB in their urine. This is not surprising, given that personal care products, which often contain these preservatives, are a primary route of exposure to chemicals associated with chronic health conditions. Unfortunately, there are few health programs that offer this type of education, and the medical community, which could be a useful vector for such information, also remains largely unaware of these environmental health issues [16,21].

Interestingly, people in better health had higher levels of MECPP and MBP. As described above, even products labeled “phthalate free”, which may be used by people aiming to improve their health, can still be contaminated with phthalates—both intentionally added (such as in fragrances) and through contamination (such as in packaging). Further, people who consider themselves healthy because they exercise often can be exposed to a variety of harmful chemicals in gymnasiums and sports equipment [74,75,76,77,78,79], and phthalates can be present in products such as athletic clothing, yoga mats, and flooring.

Despite this study being conducted under the context of “real world” situations vs. well-controlled research settings, one limitation of this research is that underreporting is common. People rarely include every product they use, especially when self-reporting through a mobile app. Also, we focused this study on personal care, household products, and supplements, while there are many other sources of everyday chemical exposure. These include household items such as furniture and electronics, environmental exposures such as water, food, and house dust, and occupational exposures. In our current research, we are improving our documentation of these sources and studying a larger sample of participants to increase our ability to identify important real-world relationships between environmental exposures and harmful chemicals detected in urine.

This study provides support for our hypothesis that the number of products a person uses is associated with urinary metabolites of commonly used intentionally and non-intentionally added chemicals that are typically not disclosed on product labels. We chose to study personal care and cleaning products because these products often contain transient toxic chemicals, such as parabens and phthalates, that may be eliminated from our bodies relatively quickly. Therefore, identifying products that are causing high exposures to toxic chemicals can make reducing these exposures actionable. It is unfortunate that the burden falls on the consumer to understand and avoid such exposures, but until better regulation is in place to ban these chemicals from products, the empowerment of individuals remains an important tool.

## 5. Conclusions

Greater product use and more ingredients of concern are associated with urinary metabolites of known EDCs and self-ratings of poor health. Women and people who take supplements are at greater risk, and even people who consider themselves to be healthy can be highly exposed. Products that may be advertised as free of these chemicals may still be contaminated unknowingly through packaging or production. Lifestyle interventions and educating the general public can make people aware of the presence of these chemicals in their everyday products so they can make efforts to avoid them. This can be achieved through digital health interventions and educational tools [80]. Researchers are working to build these interventions and improve environmental health literacy.

## Figures and Tables

**Figure 1 ijerph-22-00099-f001:**
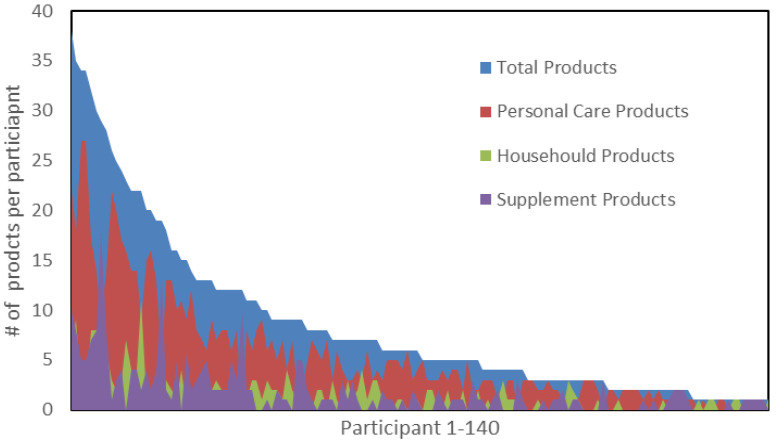
Number of products reported by each participant.

**Figure 2 ijerph-22-00099-f002:**
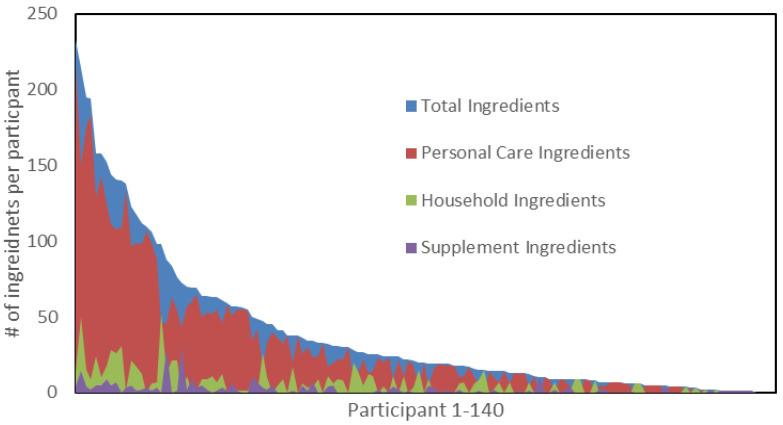
Number of ingredients of concern identified for each participant.

**Figure 3 ijerph-22-00099-f003:**
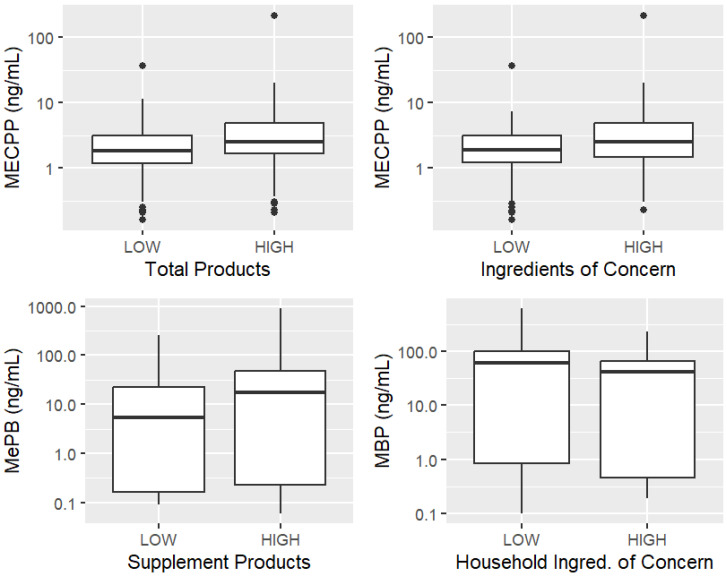
Box plots of statistically discernible comparisons of the number of products and ingredients of concern by urinary metabolites.

**Table 1 ijerph-22-00099-t001:** Demographics of 140 participants.

Demographics ^a^		
Age in Years		31 (21, 40)
Body Mass Index		26 (17, 53)
Sex	Male	25 (18%)
	Female	115 (82%)
Race	White	114 (81%)
	Hispanic	9 (6%)
	Asian	7 (5%)
	Native American, Pacific Islander, Black, Other	10 (7%)
Income	Less than USD 50,000	28 (26%)
	USD 50,000 to USD 100,000	48 (45%)
	More than USD 100,000	30 (28%)
	Missing	34
Education	Some college or Less	20 (19%)
	Associate Degree	11 (10%)
	Bachelor’s Degree	45 (42%)
	Master’s Degree or Higher	31 (29%)
	Missing	33
Self-Reported Health Status	Poor/Fair	9 (8%)
	Good	33 (31%)
	Very Good	53 (50%)
	Excellent	12 (11%)
	Missing	33

^a^ For categorical variables, counts (frequencies) and percentages are reported; percentages are taken over non-missing values only. For continuous variables, means and ranges (i.e., min, max) are reported.

**Table 2 ijerph-22-00099-t002:** Number of products reported and ingredients of concern identified for 140 participants.

	Number of Products			Number of Ingredients of Concern		
	Min–Max	Median	Mean (SD)	Min–Max	Median	Mean (SD)
Total Products	1–38	5.5	8.49 (8.38)	0–233	19.5	38.89 (48.1)
Personal Care	0–27	4	6.13 (5.57)	0–215	19.5	36.17 (42.92)
Household	0–11	2	2.35 (1.93)	0–52	7.5	9.59 (9.63)
Supplement	0–18	2	2.81 (2.88)	0–30	2	3.13 (4.86)

**Table 3 ijerph-22-00099-t003:** Demographics by dichotomized (Low/High) number of products and ingredients of concern.

Demographics		Number of Products			Number of Ingredients of Concern		
		Low (n = 70)	High (n = 70)	Test Statistic ^a^	Low (n = 70)	High (n = 70)	Test Statistic ^a^
Age (mean, standard deviation)		32 (4.52)	31 (4.37)	U = 2146.5 *p* = 0.257	31 (4.520)	31 (4.4)	U = 2302 *p* = 0.633
Body Mass Index (mean, standard deviation)		27 (5.3)	25 (6.47)	**U = 1045.5** ** *p* ** ** = 0.018**	26 (5.33)	25 (6.56)	U = 1180 *p* = 0.118
Sex	Male (n = 25)	68%	32%	**χ** **^2^ = 3.94** ** *p* ** ** = 0.047**	68%	32%	**χ** **^2^ = 3.94** ** *p* ** ** = 0.047**
	Female (n = 115)	46%	54%		46%	54%	
Race	White (n = 114)	48%	52%	χ^2^ = 0.76 *p* = 0.385	48%	52%	χ^2^ = 0.76 *p* = 0.385
	Non-White (n = 26)	58%	42%		58%	42%	
Income	Less than USD 50,000 (n = 28)	36%	64%	H = 2.54 *p* = 0.281	50%	50%	H = 0.095 *p* = 0.954
	USD 50,000 to USD 100,000 (n = 48)	46%	54%		50%	50%	
	More than USD 100,000 (n = 30)	57%	43%		47%	53%	
Education	Some college or Less (n = 20)	50%	50%	H = 1.86 *p* = 0.602	65%	35%	H = 2.70 *p* = 0.44
	Associate Degree (n = 11)	27%	73%		55%	46%	
	Bachelor’s Degree (n = 45)	49%	51%		44%	56%	
	Master’s Degree or Higher (n = 31)	48%	52%		45%	55%	
Self-Reported Health Status	Poor/Fair (n = 9)	11%	89%	H = 5.34 *p* = 0.148	22%	78%	H = 3.24 *p* = 0.356
	Good (n = 33)	52%	49%		49%	52%	
	Very Good (n = 53)	51%	49%		55%	45%	
	Excellent (n = 12)	42%	58%		50%	51%	

^a^ Test statistics: χ^2^ = Pearson chi-square, U = Mann–Whitney statistic, H = Kruskal–Wallis statistic. Bold text denotes associations with *p*-values < 0.05. Rows that do not add to 100% are due to rounding to whole numbers.

**Table 4 ijerph-22-00099-t004:** Means (ng/mL) of urinary metabolite quartiles.

Urinary Metabolites	LOD ^a^	% above LOD ^a^	Q1	Q2	Q3	Q4
Methylparaben (MePB)	0.25	64%	0.12	2.35	17.62	188.21
Propylparaben (PPB)	0.25	41%	0.10	0.15	1.64	66.78
Monobutyl phthalate (MBP)	0.50	79%	0.33	33.87	62.87	194.76
Monoethyl phthalate (MEP)	0.60	69%	0.29	10.14	56.00	538.91
Mono-(-ethyl-5-carboxypentyl) phthalate (MECPP)	0.50	91%	0.73	1.78	2.89	13.51

^a^ LOD = Level of detection.

**Table 5 ijerph-22-00099-t005:** Associations between the dichotomized (Low/High) number of products or ingredients of concern and urinary metabolite measurements.

	Mann–Whitney Statistic	*p*-Value
Mono-(-ethyl-5-carboxypentyl) phthalate (MECPP)		
Total Products	U = 3027.50	**0.013**
Total Ingredients of Concern	U = 2853	0.083
Personal Care Products	U = 2373.5	**0.001**
Personal Care Ingredients of Concern	U = 2198.5	**0.031**
Methylparaben (MePB)		
Supplement Products	U = 1229.5	**0.03**
Monobutyl phthalate (MBP)		
Household Ingredients of Concern	U = 721	0.069

Bold text denotes associations with *p*-values < 0.05.

## Data Availability

Data are available upon request from the corresponding author.

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
