# Peer review of "Associations Between Daily-Use Products and Urinary Biomarkers of Endocrine-Disrupting Chemicals in Adults of Reproductive Age"

_ijerph, 2025, doi:10.3390/ijerph22010099_

Round 1

Reviewer 1 Report

Comments and Suggestions for Authors

I appreciate the opportunity to review this exciting manuscript, which provides valuable insights into how individual exposure to endocrine-disrupting chemicals (EDCs) in everyday products may affect human health. The article is well-written, scientifically engaging, and aligns closely with the International Journal of Environmental Research and Public Health (IJERPH) aims. However, before it reaches its full potential for publication, I recommend addressing a few critical aspects, as outlined below. 

Specific Comments for Improvement: 

1. Clarification of Participant Recruitment in Section 2.1:

The manuscript mentions that participants were invited via email but did not specify the criteria for selecting these individuals. For greater transparency, it would be beneficial if the authors could elaborate on the following:

* The criteria used to determine eligibility for participants whose emails were sent will help the readers understand the selection process and the sample's representativeness.

* The origin of the email list, whether institution-based or a public health registry, to provide a clear understanding of the source of participants.

This information will help readers better understand the recruitment methodology, enhance the study's rigor and reproducibility, and instill confidence in the research process.

2. Description of analytical methods (LC/MS/MS):

While the authors indicate using liquid chromatography-tandem mass spectrometry (LC/MS/MS) for analysis, a brief description of this technique, especially detailing any specific adaptations made for this study (including chromatographic conditions), would enhance the methodology section. 

3. Addition of Data Visualization in Results Section 3.1:

In Section 3.1, where results are discussed, a table summarizing the main data points would be beneficial. Adding a well-organized table will allow readers to observe and interpret the data referenced in the text easily. 

4. Clarification of the terminology and accuracy in Table 2:

* In Table 2, the term "BIPOC" might not be universally understood by international readers, as it is primarily a US-based term. To make the manuscript more accessible, consider defining "BIPOC" (Black, Indigenous, and People of Color) in the table or using a more universally understood term.

* Some of the percentages in Table 2 do not sum up to 100%, possibly due to rounding errors or other inconsistencies. These errors could potentially mislead readers and affect the accuracy of the representation of the demographics of the participants. Therefore, reviewing these values for accuracy is essential to ensure clarity and precision. 

5. Streamlining Table 2 for readability:

To enhance readability, consider removing the chi-square, H, and U values from Table 2 and retaining only the p-values. Additionally, include the type of analysis performed as a footnote. 

6. Footnote Quartile Definitions (Q1, Q2, Q3, Q4):

Using quartiles (Q1, Q2, Q3, Q4) is an effective method for categorizing data. However, defining these terms in a footnote or within the text of the relevant table would be helpful for readers. Although these terms are commonly used in statistics, a simple explanation of their meaning in this context will enhance clarity. 

7. Improvement of Discussion Content (Lines 370-374):

Lines 370-374 introduce an inappropriate discussion referencing a specific private company. This inclusion could detract from the study's objectivity and focus. It is recommended that this segment be removed or revised to maintain a neutral tone, avoiding any mention of the company by name.

Overall assessment:

With these minor adjustments, this article has the potential to make a valuable contribution to the International Journal of Environmental Research and Public Health (IJERPH). The strengths include its thorough analysis and significant implications for public health. Addressing the abovementioned suggestions will improve the article's readability, transparency, and accessibility for a wider audience. Once revised, I am confident it will be well-suited for publication.

Thank you for considering my feedback.

Author Response

Thank you for your kind assessment and thoughtful comments to improve the manuscript.

1) Clarification of Participant Recruitment in Section 2.1:

Participants were recruited from The Healthy Nevada Project, a community health research program. Participants in the program who agreed to be reconnected for future research opportunities, met the age criteria and showed interest in future studies were recontacted and emailed information about the current study. This information has been added to section 2.1 of the manuscript.

2) Description of analytical methods (LC/MS/MS): While the authors indicate using liquid chromatography-tandem mass spectrometry (LC/MS/MS) for analysis, a brief description of this technique, especially detailing any specific adaptations made for this study (including chromatographic conditions), would enhance the methodology section.

 The details of the method have been published in a previous paper [43]. In brief, 100 μL of urine was combined with 100 μL of water, isotopically labeled standards were added to both the samples and blank water, and cocktail standards were spiked at 5 μL each. The samples and standards were incubated with 25 μL of β-Glucuronidase buffer for 2 hours at 37 °C, followed by the addition of 275 μL of water to the vials. Solvent blanks were prepared simultaneously. Samples were injected in duplicate, with blanks inserted after each duplicate. The analysis was conducted using an Agilent Prochell 120-EC18 column (Agilent Technologies, Santa Clara, CA, USA). See page 4, lines 160-167 

3) Addition of Data Visualization in Results Section 3.1: In Section 3.1, where results are discussed, a table summarizing the main data points would be beneficial. Adding a well-organized table will allow readers to observe and interpret the data referenced in the text easily. 

An additional demographics table (Table 1) has been added (page 5-6, lines 242-260). 

4) Clarification of the terminology and accuracy in Table 2:

* In Table 2, the term "BIPOC" might not be universally understood by international readers, as it is primarily a US-based term. To make the manuscript more accessible, consider defining "BIPOC" (Black, Indigenous, and People of Color) in the table or using a more universally understood term.

* Some of the percentages in Table 2 do not sum up to 100%, possibly due to rounding errors or other inconsistencies. These errors could potentially mislead readers and affect the accuracy of the representation of the demographics of the participants. Therefore, reviewing these values for accuracy is essential to ensure clarity and precision. 

 The term has been changed in Table 2 to “Non-White." For the percentages in Table 2 (now Table 3), one was an error, the rest was due to rounding, which has been resolved.

5) Streamlining Table 2 for readability: To enhance readability, consider removing the chi-square, H, and U values from Table 2 and retaining only the p-values. Additionally, include the type of analysis performed as a footnote.

Because all statistical tests were not the same, we think it is best to keep them in the table to make it clear to the reader that different tests were run. The footnote explaining the test is at the end of table 3. 

6) Footnote Quartile Definitions (Q1, Q2, Q3, Q4): Using quartiles (Q1, Q2, Q3, Q4) is an effective method for categorizing data. However, defining these terms in a footnote or within the text of the relevant table would be helpful for readers. Although these terms are commonly used in statistics, a simple explanation of their meaning in this context will enhance clarity. 

We added the definition (page 10, lines 335-337). An additional reference with NHANES analysis methodology (similar to ours) has been added. 

7. Improvement of Discussion Content (Lines 370-374): Lines 370-374 introduce an inappropriate discussion referencing a specific private company. This inclusion could detract from the study's objectivity and focus. It is recommended that this segment be removed or revised to maintain a neutral tone, avoiding any mention of the company by name.

The reference has been removed.

Reviewer 2 Report

Comments and Suggestions for Authors

This study well designed and executed so well. There are no queries but few suggestions given below, the authors can respond to these suggestions

1.      1 The title can be rephrased little more interesting so that the  common readers will be interested to read the article (suggestion only)

2.      2 Instead of 24 hrs study the authors could have extended short term and long term exposure as well, so that the secondary metabolites or disintegration of any phenolic EDCs could have been recorded.

3.      3 Could you find any control participant? Who has not exposed to any of these EDCs?

4.      4 Could you find and correlate the initials known concentration applied on skin and how much excreted through urine?

5.      5 Are these products used here in this studies  passed FDA QC checked approval?

6.      6 Could you please include the questionnaire as an annexure or attachment to this article ?

7.      7 Have you followed any specific inclusion or exclusion criteria other than age?

8.      8 Have you noticed any shocking or interesting findings in this study?

9.      9 This study will attract many young researchers to design the same study with their native participants hence the methodology part could have been explained well in this article, the sample processing, EDC extraction, LC/MC conditions and mobile phase ratio, standard run, trouble shooting etc..the authors have cited “38” as reference but they have not explained the methodology as well.

110 In the conclusion part, authors can add few bulleted points to the readers as caution/warning/awareness message.

1 11 Without disclosing the brand name, the authors could have mentioned these EDCs from sunscreen or moisturizer or lotion or any other make up/cosmetics.

1 12 Over-all this article is very interesting and has a message to the society

Author Response

Thank you for your kind assessment and thoughtful comments to improve the manuscript.

1) The title can be rephrased little more interesting so that the  common readers will be interested to read the article (suggestion only)

Thank you for your suggestion. After consideration, we have decided to keep the current title.

2) Instead of 24 hrs study the authors could have extended short term and long term exposure as well, so that the secondary metabolites or disintegration of any phenolic EDCs could have been recorded.

 We will consider this analysis in the future. 

3) Could you find any control participant? Who has not exposed to any of these EDCs?

 Controls for this would be difficult as the majority of the population (>95% for many chemicals) has EDC exposure. 

4) Could you find and correlate the initials known concentration applied on skin and how much excreted through urine?

Since this was not a controlled study, we don’t have this information.

5) Are these products used here in this studies  passed FDA QC checked approval?

No, the FDA does not approve personal care products, but it does have regulatory authority over ingredients used in some products (e.g. cosmetics).

6) Could you please include the questionnaire as an annexure or attachment to this article?

 The survey asked participants to list their diet and products used from the previous 24 hours, so there are no specific questions we can list.

7) Have you followed any specific inclusion or exclusion criteria other than age?

This is reported in Section 2, Lines 134-137: “Those who responded were required to meet the following eligibility criteria:  be between the ages of 18 and 40, own a smartphone, speak English, not be pregnant, and have no known diagnoses of cancer, metabolic disorders, or kidney disease.”

8) Have you noticed any shocking or interesting findings in this study?

 We found that products might be contaminated through plastic packaging, during the manufacturing process or possibly adulterated. We also felt the association with supplement use was interesting. These are both highlighted in the discussion.

9) This study will attract many young researchers to design the same study with their native participants hence the methodology part could have been explained well in this article, the sample processing, EDC extraction, LC/MC conditions and mobile phase ratio, standard run, trouble shooting etc..the authors have cited “38” as reference but they have not explained the methodology as well.

We have expanded the methods section. Please See page 4, lines 160-167.  

10) In the conclusion part, authors can add few bulleted points to the readers as caution/warning/awareness message.

The conclusion was expanded. See page 14, lines 473-477. 

11) Without disclosing the brand name, the authors could have mentioned these EDCs from sunscreen or moisturizer or lotion or any other make up/cosmetics.

Please see lines 414-421 for additional information on specific EDCs. 

12) Over-all this article is very interesting and has a message to the society.

Thank you!

Reviewer 3 Report

Comments and Suggestions for Authors

This pilot study explores concentrations of EDCs in urine samples and products used 24 hours prior to sample collect and ingredients of concern in products mentioned (140 adults of reproductive age in Northern Nevada). Overall, study shows merit and significance for public health and environmental health sciences. Suggestions include rewriting parts of study to avoid iThenticate overlap (match overview 1-3 that is above 1%).

Author Response

This pilot study explores concentrations of EDCs in urine samples and products used 24 hours prior to sample collect and ingredients of concern in products mentioned (140 adults of reproductive age in Northern Nevada). Overall, study shows merit and significance for public health and environmental health sciences. Suggestions include rewriting parts of study to avoid iThenticate overlap (match overview 1-3 that is above 1%).

Thank you for the suggestion, with edits from the reviewers significant parts have been rewritten. I believe this will address the issue.